# Measuring Spatial and Temporal PM_2.5_ Variations in Sacramento, California, Communities Using a Network of Low-Cost Sensors

**DOI:** 10.3390/s19214701

**Published:** 2019-10-29

**Authors:** Anondo Mukherjee, Steven G. Brown, Michael C. McCarthy, Nathan R. Pavlovic, Levi G. Stanton, Janice Lam Snyder, Stephen D’Andrea, Hilary R. Hafner

**Affiliations:** 1Sonoma Technology, 1450 N. McDowell Blvd., Suite 200, Petaluma, CA 94954, USA; amukherjee@sonomatech.com (A.M.); mmccarthy@sonomatech.com (M.C.M.); npavlovic@sonomatech.com (N.R.P.); lstanton@sonomatech.com (L.G.S.); hilary@sonomatech.com (H.R.H.); 2Department of Atmospheric and Oceanic Sciences, University of Colorado Boulder, Boulder, CO 80309, USA; 3Sacramento Metropolitan Air Quality Management District (SMAQMD), Sacramento, CA 95814, USA; jlam@airquality.org (J.L.S.); sdandrea@airquality.org (S.D.)

**Keywords:** low-cost sensor, particulate matter, air quality, calibration strategies, network design

## Abstract

Low-cost sensors can provide insight on the spatio-temporal variability of air pollution, provided that sufficient efforts are made to ensure data quality. Here, 19 AirBeam particulate matter (PM) sensors were deployed from December 2016 to January 2017 to determine the spatial variability of PM_2.5_ in Sacramento, California. Prior to, and after, the study, the 19 sensors were deployed and collocated at a regulatory air monitoring site. The sensors demonstrated a high degree of precision during all collocated measurement periods (Pearson R^2^ = 0.98 − 0.99 across all sensors), with little drift. A sensor-specific correction factor was developed such that each sensor reported a comparable value. Sensors had a moderate degree of correlation with regulatory monitors during the study (R^2^ = 0.60 − 0.68 at two sites). In a multi-linear regression model, the deviation between sensor and reference measurements of PM_2.5_ had the highest correlation with dew point and relative humidity. Sensor measurements were used to estimate the PM_2.5_ spatial variability, finding an average pairwise coefficient of divergence of 0.22 and a range of 0.14 to 0.33, indicating mostly homogeneous distributions. No significant difference in the average sensor PM concentrations between environmental justice (EJ) and non-EJ communities (*p* value = 0.24) was observed.

## 1. Introduction

In the United States, the Environmental Protection Agency (EPA) regulates the ambient concentrations of various air pollutants, including particulate matter (PM) with an aerodynamic diameter of 2.5 µm and smaller (PM_2.5_, also referred to as fine particulate matter). The adverse impacts of PM on human health depends on the size distribution because fine particles can travel more deeply through the respiratory system than larger particles [1]. EPA has established National Ambient Air Quality Standards (NAAQS) to regulate the concentrations of PM_2.5_ and other pollutants in order to minimize health and environmental impacts [2]. In order to ensure these standards are met, federal, state, local, and tribal environmental agencies operate instruments that collect data for regulatory purposes that meet rigorous quality standards. Regulatory instruments that meet these standards are designated as Federal Reference or Equivalent Methods (FRM or FEM) by the EPA, provided they are calibrated and operated according to standardized procedures [3,4].

The network of regulatory air quality monitors was designed to monitor regional compliance with federal and state health standards. FRM or FEM particulate instruments are typically large and expensive ($20,000 or more), demand substantial power, and require periodic maintenance and/or calibration by trained operators. Many field studies have shown that within given urban regions there can be significant spatial variability of PM at small spatial scales, driven by heterogeneous local emissions, local topography, and the microscale variability of meteorology conditions [5,6,7,8]. For example, Superczynski and Christopher [9] found that the annual average PM_2.5_ varied by 25–30% across the Birmingham, Alabama, in 2000–2009. Recent studies have examined the spatial variability of PM using networks of sensors [6,8,10].

Many low-cost sensor models cost only a few hundred dollars and are marketed directly to consumers who are interested in understanding personal exposure or citizen science. These sensors are designed to be simple to operate, with little power or infrastructure needed compared to regulatory grade monitors. Low-cost sensors have the potential to provide valuable information on the spatial and temporal variability of PM and other pollutants, and if sufficient, steps are taken to ensure that the quality of the measurements is robust enough to meet the given objectives [11,12].

Validating sensor measurements, and in particular, determining the comparability among sensors in an ambient environment, is a critical step to ensure data quality for any sensor deployment. The EPA’s Air Sensor Guidebook states that “Sensor calibration is vital for producing accurate and usable data” [13]. Recent studies have assessed the performance capabilities of sensors in the laboratory and in field conditions. Studies have used sensor measurements to examine air quality and demonstrate the value of sensor measurements in assessing spatio-temporal variability when a careful procedure to evaluate sensor measurements has been employed. Snyder et al. [14] provide an overview of the new generation of low-cost sensors that measure gas phase and particulate air pollution. Previous studies have utilized measurements from these sensors to examine the impact of air pollution exposure on health [15,16] and to examine their potential for integration into a network of stations [6,7]. Gao, Cao and Seto [6] deployed a network of eight sensors to assess the spatial variability of PM in Xi’an, China, and to identify air pollution hot spots, after calibrating to an established gravimetric method. Heimann, et al. [17] utilized a network of carbon monoxide sensors to assess the source apportionment of emission drivers. The EPA-conducted Community Air Sensor Network (CAIRSENSE) project assessed the performance of gas phase and particulate matter sensors over an eight-month period and quantified the unique precision and accuracy of different sensors, compared to FEM monitors [18]. Zikova et al. [8] utilized 27 PM sensors to evaluate the spatial variability of PM across Rochester, NY. There is interest in synthesizing measurements from sensors along with the existing regulatory air quality infrastructure to examine exposure and air pollution dynamics at finer scales [18,19,20].

The sensor model used in this study is the AirBeam, with a cost of USD $250, which was developed by HabitatMap, Inc., in Brooklyn, NY, to measure PM_2.5_ using a well-established optical technique. In previous studies, the AirBeam was shown to demonstrate high sensor-to-sensor precision in the environments of Decatur, Georgia, and the Cuyama Valley of California [18,21]. These studies also examined the performance of the AirBeam relative to reference regulatory monitors and showed that meteorological variables such as relative humidity and wind speed had predictive capacity with regard to the deviation between AirBeam and reference instruments. The Air Quality Sensor Performance Evaluation Center (AQ-SPEC) has assessed the AirBeam and multiple other sensors in laboratory and field conditions. In laboratory comparisons against the GRIMM FEM monitor, the AirBeam sensors showed good correlation (R^2^ ~ 0.87) at 5-minute resolution, while underestimating PM_2.5_ by approximately a factor of 5 [22]. Similar results were demonstrated in the field, and raw particle counts had better agreement than retrieved PM_2.5_ concentrations between the two instruments [23]. AQ-SPEC assessments also show that the AirBeam has high precision over a range of temperature and relative humidity conditions.

Previous studies have used emission inventories to examine the spatial variability of pollution sources [24,25,26]. In this study, the measurements from a network of AirBeam sensors are compared to a gridded emissions inventory to examine whether emission hotspots result in elevated PM_2.5_ concentrations detectable by sensors. While directly comparing emissions to concentrations neglects processes, such as secondary aerosol formation, and provides a way to examine the spatial variability of emissions and concentrations similar to methods used by Mohan et al. [24].

In this study, AirBeam sensors were deployed in the urban wintertime environment of Sacramento, California, as part of a larger study to understand the differences in air pollution among environmental justice (EJ) and non-EJ communities [27]. Here, we report on the sensor-to-sensor variability during collocated measurement periods, the accuracy of the sensors in comparison to FEM and FRM measurements at regulatory monitoring sites, the spatio-temporal variability of PM_2.5_ during winter 2016/2017 among EJ and non-EJ communities, and the relationship between emissions and measured PM_2.5_.

## 2. Methods

### 2.1. Instrumentation

The AirBeam sensor was used for measuring PM_2.5_ at community sites. HabitatMap, Inc., has developed the AirBeam on an open source platform, with details and schematics available at http://aircasting.org/about. The AirBeam measures PM_2.5_ using an optical technique. A light-emitting diode (LED) source of visible green light is used to detect particles, and the raw measurement provides total particle counts. This particle count is then converted to mass concentration, assuming a fixed size distribution and constant particle density; the October 2016 version of software provided by HabitatMap was used in this study. This default retrieval algorithm was: PM_2.5_ (µg m^−3^) = 0.518 + 0.00274 × particle count (hppcf). Field evaluations have shown that the AirBeam size cutoff and assumed size distribution provides a fairly accurate representative sample of PM_2.5_ (R^2^ of approximately 0.75) in comparison to GRIMM derived PM_2.5_ [21]. An EPA study examined the performance of 12-hour average measurements from the AirBeam compared to a regulatory monitor and found high precision (sensor-to-sensor R^2^ = 0.99) and moderate accuracy (sensor-to-reference R^2^ = 0.43), with modest improvements from meteorology corrections (sensor-to-reference R^2^_adj_ = 0.51) [18]. 

In this study, AirBeams were mounted on tripods covered in a metal hood to repel rain, but not to obstruct air flow to the AirBeam. From the manufacturer, AirBeams are outfitted with Bluetooth technology to transfer data from the sensor to a handheld smart device or data storage location. Due to the limited range of Bluetooth (approximately 5–10 ft (1.5–3 m)), AirBeams used in the study were configured with Valarm serial adaptors and wireless cellular network components to send data in real-time from the AirBeam to Valarm’s data cloud; the data were then retrieved every minute and stored in a central database. Valarm’s Yoctopuce Yocto serial sensor adaptors were connected to the AirBeam’s auxiliary ports via single-pin wires, which gathered data from the AirBeam for delivery. Connected to the serial adaptor via micro-USB was Valarm’s GSM 3G sensor hub, outfitted with a Ting GSM SIM card, which transmitted data to Valarm’s cloud. Sonoma Technology, Inc., then retrieved these data from Valarm’s cloud API in real time. One-minute AirBeam sensor data were averaged to hourly values, requiring 75% completeness for each hour.

Two different methods were used to power the AirBeam units used in the study: Hardwired AC power, and a battery-solar power combination. The power method for each AirBeam monitoring unit was determined on a site-by-site basis depending on available resources. If the hosting site contained an accessible outdoor power outlet (with permission from the owner), a heavy-duty outdoor-rated extension cable was installed inside the AirBeam’s utility box as a direct power source. A majority of the AirBeam monitoring units were powered by this method. If the hosting site did not have an accessible outdoor power outlet, a battery-solar power configuration was used. Due to the limited power needs of the AirBeam monitoring unit, a 12-volt deep cycle marine battery was wired to the AirBeam’s utility box and served as the main power source for the unit. Inside the AirBeam’s utility box, a Valarm 12-volt DC power regulator with a micro-USB port was wired to the battery cables and served as a platform to connect the AirBeam and data communication components. In addition to the battery, a solar panel was mounted to the top of the AirBeam’s monitoring unit and wired into the battery’s wiring harness with a solar controller regulator to charge the battery. The AirBeams and Valarm data communication components are connected to a power source via USB. Both of these methods, hardwire and solar-battery, yielded a connection for a multi-port USB adaptor, enabling power sources for all devices.

Measurements from regulatory monitors were reported to and accessed from the EPA’s Air Quality System (AQS, available at https://aqs.epa.gov/api). A Met One Instruments Model 1020 Beta Attenuation Monitor (BAM) and a daily FRM filter sampler were operated by the Sacramento Metropolitan Air Quality Management District (SMAQMD) at the Sacramento-Del Paso Manor air monitoring site (Del Paso Manor, AQS site code 06-067-0006) and by the California Air Resources Board (ARB) at the Sacramento 1309 T Street air monitoring site (T Street, AQS site code 06-067-0010) during the study. The BAM measures PM_2.5_ on an hourly basis by sampling ambient air through a sharp cut cyclone and depositing particles on a filter tape, and then exposing the filter tape to a source of beta radiation. The FRM R & P Model 2025 PM_2.5_ Sequential Air Sampler provided daily average measurements of PM_2.5_ using a gravimetric filter sampling technique. 

Meteorological measurements were also collected at Del Paso Manor and T Street by SMAQMD and ARB. Hourly measurements of temperature, relative humidity, and wind speed and direction were measured and accessed through AQS. Dew point was calculated from measurements of temperature and relative humidity using the Clausius-Clapeyron equation:

Dew Point = (237.3 log(Vapor Pressure) − 186.52)/(8.29 − log(Vapor Pressure)).
(1)

### 2.2. Study Design

Sacramento is located in the southern portion of the Sacramento Valley, an urbanized area with flat topography approximately 100 km from the Pacific Ocean. Source apportionment studies show that in the Sacramento Valley, the major chemical components of wintertime PM_2.5_ are ammonium nitrate and organic carbon [25,26]. Combustion from gasoline and diesel engines contributes to the majority of nitrate mass, whereas agricultural sources are the major source of secondary ammonium ion [26]. In Sacramento, the organic carbon and elemental carbon components of PM are driven by emissions from diesel, gasoline, and wood burning [27,28].

The AirBeams were deployed in three distinct phases for the study:
(1)A pre-study collocation period (11/10/16–11/16/16) where 19 AirBeams were collocated with the BAM, FRM, and meteorology measurements at the Del Paso Manor site.(2)The study period (2 months, 12/1/2016–02/1/2017), during which 19 AirBeams were deployed at 15 locations in Sacramento. Three AirBeams were collocated at both Del Paso Manor and T Street sites with the BAM and FRM monitor, in order to assess sensor precision and drift during the study period. The remaining 13 AirBeams were deployed individually at site locations as shown in Figure 1.(3)A post-study collocation period (2/4/2017–3/8/2017) where 19 AirBeams were collocated with the BAM, FRM and meteorology measurements at the Del Paso Manor site in the same configuration as in the pre-study collocation period.

Figure 1 shows the community site locations and the Del Paso Manor and T Street regulatory monitoring sites where the sensors were deployed during December-January. Based on the EPA’s EJScreen EJ Index data (available at https://www.epa.gov/ejscreen/download-ejscreen-data) for particulate matter, three EJ and three non-EJ community areas were selected. In each community area, between one and four community sites were selected. These sites were located at private residences whose owners volunteered the use of their property for the study. The study domain includes a 14 km by 16 km, 224 km^2^ area in Sacramento, with all sites situated in an urban environment.

#### 2.2.1. Collocation and AirBeam Correction

Studies have shown that factory default sensor measurements can have sensor-sensor offsets that can be corrected for [8,10,20,28]. This correction can be developed from an independent reference instrument or from a set of sensors of the same model. In this study, the measurements from pre- and post-study collocation periods were used to develop the correction for each AirBeam, quantify precision (AirBeam versus AirBeam variability), and assess whether the response of individual AirBeams to ambient PM_2.5_ concentrations drifted during the study. The Pearson coefficient of determination (R^2^) was computed between each sensor, and the 19-sensor mean to understand precision and sensor-to-sensor variability. Linear regressions between each sensor and the 19-sensor mean were computed to calculate the correction needed for each sensor. The regressions for each sensor were calculated and compared for the pre- and post-study collocation periods, to determine whether the offset of each sensor was stable and consistent during the study. These linear regressions were used to compute a correction factor for each sensor using the equation:
Corrected AirBeam = (Raw AirBeam − Intercept)/Slope
(2)

The correction ensures that, when the sensors are deployed as a network during the study period, each sensor provides an intercomparable value. All data shown in the Results section are corrected.

Collocated sensor measurements were used during the study period to examine precision. The degree of confidence in sensor precision is presented statistically as the standard deviation of the difference between the linear regression model and the 19-AirBeam average value. Evaluating precision during these periods allows us to quantify how much of the variability from the study period was due to varying aerosol conditions versus sensor-to-sensor variability.

The data from the collocation of AirBeam sensors, BAMs, and meteorological measurements at Del Paso Manor and T Street, during the study, were used to quantify drift, accuracy, and the influence of meteorological conditions on the corrected AirBeam measurements. The relationship between corrected AirBeam measurements and BAM measurements was examined at Del Paso throughout the pre-study, study, and post-study periods to assess the stability of the AirBeam measurements. The correlation between corrected AirBeam and BAM measurements was used to assess accuracy. Daily averages of AirBeam measurements were also compared with the daily FRM measurements. The role of meteorological influence on sensor accuracy was assessed at both sites, using multivariate linear regression of AirBeam data with meteorological variables against BAM measurements. This analysis provides insight into the performance of the AirBeam instrument given its measurement technique and the impact of varying meteorological conditions in the Sacramento wintertime environment.

#### 2.2.2. Calculations from AirBeam Deployment in Communities

Corrected AirBeam measurements are used to provide estimates of the spatial and temporal variability of PM_2.5_ during the study period. The variability of PM_2.5_ between EJ and non-EJ sites is estimated through examining the distributions of AirBeam PM_2.5_ measurements and using a Student’s t-test of the sample means. The R^2^ and the coefficient of divergence (COD, defined below) calculated between AirBeam-AirBeam pairs and AirBeam-BAM measurements are used to evaluate spatial heterogeneity of measurements during the study. The pairwise COD for sites j and k is defined as:(3)$$COD_{jk}=\sqrt{\frac{1}{p}\sum_{i=1}^{p}{\left(\frac{x_{ij}-x_{ik}}{x_{ij}+x_{ik}}\right)}^{2}}.$$
where *i* is the *i*th measurement out a total of *p* measurements, and *j* and *k* are the two sites being compared. COD values range from 0 to 1, with a value of 0 indicating identical measurements, a value of 0.2 being a threshold of heterogeneous spatial distributions and a value of 1, indicating significant heterogeneity [29,30]. COD analysis has been used to examine the spatial variability of air pollutants in similar special studies [8,30,31].

To assess the relationship between emissions inventories and ambient concentrations, the measurements were compared to the wintertime PM_2.5_ emissions inventory (EI) for the study area. SMAQMD provided 2012 PM_2.5_ EI data for weekdays and weekends in the winter season on a 4-by-4 km grid to the California Air Resources Board; these were processed through the California Emissions Projection Analysis Model (CEPAM): External Adjustment Reporting Tool. The inventory provides PM_2.5_ emissions data in tons emitted per day in each grid cell. These grid cell values were compared to the PM_2.5_ increment above background measured by all AirBeams located within the grid cells. Emissions data were compared to the average of all AirBeam PM_2.5_ measurements collected by all sites located in that grid cell. For each grid cell, the average concentration across all AirBeams in the grid cell was determined using only hours when all AirBeams had hourly average data, and the increment above the lowest measured value (background) was calculated. A linear regression equation was fit to the average PM_2.5_ increment and the EI data, and the R^2^ was calculated for the regression.

## 3. Results

### 3.1. Precision, Correction, and Drift

Sensor precision and bias were assessed using collocated measurement periods before, during, and after the study. Raw, uncorrected data are presented in this section; subsequent sections use corrected data. The sensors demonstrated high precision and consistent measurements throughout all periods, allowing the development of a correction, as shown in Table 1. The R^2^ value of each individual AirBeam compared to the 19-AirBeam hourly mean provides a representation of sensor precision. The AirBeams demonstrate very high precision during all collocation periods, with these R^2^ values ranging from 0.98 to 0.999. This indicates that the AirBeam sensors provided stable and consistent measurements over a range of different wintertime meteorological and physical conditions, including the varying chemical composition and size distribution of PM found in Sacramento. Intercepts and slopes of regression, computed for each AirBeam versus the 19-AirBeam hourly mean for the pre-study and post-study collocation period, are presented in Table 1. With the exception of one outlier AirBeam (Henrietta), the standard deviation of the absolute value of the deviation of these slopes from the mean is 6.9%, showing how close the measurements from each AirBeam were to the 19-AirBeam average. Likewise, the intercepts of regression are of small magnitude, between −3.5 and 3 µg m^−3^. This range of linear regression variables shows that the raw AirBeam measurements are closely intercomparable and within 15% of the AirBeam mean. 

This precision is also reflected in the Root Mean Squared Error (RMSE) values. RMSE is used to quantify sensor-sensor variability, and an RMSE of 0 µg m^−3^ would indicate identical measurements. RMSE values of the raw, uncorrected data for each individual AirBeam compared to the 19-AirBeam hourly mean have an average of 1.08 µg m^−3^, maximum of 2.55 µg m^−3^ for the pre-study collocation period and a maximum of 1.52 µg m^−3^ for the post-study collocation period. The Coefficient of Variation (CV) is another metric that has been used to assess the precision of collocated sensors [32]. The average CV of all 19 AirBeams from the pre- and post-study collocation periods was found to be 0.17 ± 0.04 (average ± one standard deviation), lower than the average CV value of 0.25 ± 0.14 reported from 14 OPC-N2 PM_2.5_ measurements, reported in [32]. The high R^2^ values and low RMSE values demonstrate that the AirBeams have robust sensor-to-sensor precision.

Between the two collocation periods, the average absolute change in the slopes of regression is small (4.6%). The slopes of regression shown in Table 1 remain very consistent between the two collocation periods. In order to examine whether there was a shift between the two periods, the change in measuring 5, 10, 20, 30, 40, and 50 µg m^−3^ using the linear regressions from both periods is shown in Appendix A. The change in these values of PM_2.5_ is shown in µg m^−3^ and as a percentage. This shows the impact of the change in slopes and intercepts from the pre-study collocation period to the post-study collocation period. In a majority of cases, the change in PM_2.5_ values due to this effects is relatively negligible, being less than 5%. AirBeams measured a change of less than 10% for values between 5 and 50 μg/m^3^ in over 83% of cases. Therefore, there is high confidence that the AirBeams were measuring PM_2.5_ in a consistent way throughout the study period.

The generally robust precision of the measurements and the stable performance of the AirBeams in the pre- and post-study periods allows for a correction using the slopes and intercepts of regression against the mean. Equation (2) was applied to correct the AirBeam measurements, using the slopes and intercepts of regression presented in Table 1. Given the very similar AirBeam PM sensor response during pre-study and post-study periods, an average of the pre-study and post-study intercepts and slopes was used for the correction. For five AirBeams, the pre-study regressions alone were used for the correction factor for the study period, due to invalid data in the post-study period leading to a limited range of PM values. Because the coefficient of determination is invariant under the transformation of a linear change, the R^2^ values, shown in Table 1, are the same for the original AirBeam measurements and the corrected AirBeam measurements.

The correction allows the network of AirBeam measurements to be intercomparable throughout the study period, with each AirBeam reporting a comparable value. The corrected AirBeam measurements can then be used to assess the accuracy of the AirBeam measurements by comparing them with the BAM and FRM, as well as examining the spatial and temporal variability of PM in the Sacramento metropolitan region throughout the study period. Hereafter, the description relates to the corrected AirBeam values used in the study.

### 3.2. Collocation Results during Study 

During the study period, three AirBeams were collocated at the T Street site and three AirBeams were collocated at the Del Paso Manor site to the BAM instruments that were operating at those sites. These measurements were also examined to assess precision. A stable linear relationship was seen for these sets of AirBeams throughout the study period, consistent with Table 1. For the two sets of three AirBeams collocated during the study period, pairwise correlation R^2^ values range from 0.987 to 0.995, within the range of the pre- and post-study collocation periods. Appendix A shows an AirBeam-to-AirBeam comparison at Del Paso Manor. The standard deviation of the residuals, i.e., the difference between the linear regression model and the true AirBeam value, was computed for these six collocated AirBeams. These values range from 1.33 µg m^−3^ to 2.99 µg m^−3^, reflecting a high degree of precision. This range of PM_2.5_ values is a representation of the confidence in sensor precision during the study period. This confirms that during the study period, AirBeams demonstrated a consistent correctable bias relative to each other, and that there was no evidence of a drift in the measurement.

### 3.3. Accuracy and the Impact of Meteorology

During the study period, three AirBeams were collocated with the BAM, FRM, and meteorological measurements at Del Paso Manor. Since AirBeam measurements have been corrected at this stage, the three AirBeams present nearly identical measurements. Therefore, the results from only one AirBeam are presented. These collocated measurements provide an opportunity to examine sensor accuracy with respect to the BAM and FRM instruments, as well as to compare the AirBeam to these instruments under varying meteorological conditions. During the wintertime study period, relative humidity values were quite high, with a value of 85% or greater during 65% of the measurement hours, and 90% or greater during 57% of the time. This is reflective of the typical regional climate, with humid conditions during the winter.

Figure 2 shows a comparison of hourly corrected AirBeam values with the collocated hourly BAM measurements at Del Paso Manor during the study period, color-coded by meteorological variables that may impact the comparison between the AirBeam and BAM: (a) relative humidity, (b) dew point temperature, (c) temperature, and (d) wind speed. The R^2^ correlation of the AirBeam and the BAM is 0.60, demonstrating moderate sensor accuracy. Figure 2 indicates that there is a range of AirBeam values for a given BAM concentration, with AirBeam measurements ranging from a factor of 1 to a factor of 3 higher for the majority of measurements. In Figure 2, the deviation towards high AirBeam measurements relative to the BAM exists during the periods with high dew point and high temperature. However, under lower dew point conditions, roughly less than 4 °C, the AirBeam and BAM have approximately a 1:1 relationship (upper right plot in Figure 2). There is no apparent trend in the comparison between the measurements and either relative humidity or wind speed.

Daily average values are also important to consider because the PM_2.5_ acute NAAQS is set for daily values; therefore, the daily AirBeam values were compared to the collocated FRM. Figure 3 shows the relationship between the daily average AirBeam values and the collocated FRM values at Del Paso Manor during December 2016–January 2017. Figure 3 is analogous to Figure 2, with the same color coding used for relative humidity, dew point temperature, temperature, and wind speed. Daily average BAM measurements had a high correlation with the FRM (with R^2^ = 0.95). The R^2^ correlation of the daily average AirBeam and the FRM is 0.71. Figure 3 shows the same relationships as Figure 2, with AirBeam and FRM measurements nearly 1:1 on days with lower temperature and dew point periods. On a daily basis, the AirBeam is consistent with the FRM under low humidity conditions (i.e., with RH less than roughly 85%, dew point less than 4 °C) that were not as evident in the hourly data. No relationship is evident with wind speed.

In order to evaluate the impact of meteorological factors further, multi-variate linear regression was performed between the AirBeam measurements and the regulatory monitor measurements using meteorological variables as explanatory variables at Del Paso Manor for both hourly measurements and 24-h average values. Table 2 shows the improvement to the adjusted R^2^ values via multi-variate linear regression with meteorological explanatory variables. The improvement from each individual meteorological variable is shown, as well as the total of all four variables, and the quadratic of all four variables (including cross terms). Dew point and relative humidity are the most important explanatory variables for these cases, showing the biggest improvements in the adjusted R^2^ values for both the BAM and the FRM. This indicates that dew point and relative humidity were the most important to explain the deviation between AirBeam measurements and the regulatory monitors. 

Similar relationships, as those in Figure 2 and Figure 3, were found when the same methodology was applied to the collocated measurements at T Street. At T Street, hourly measurements collocated between the AirBeam and the BAM had an R^2^ correlation of 0.68, and daily average measurements between the same two instruments had an R^2^ correlation of 0.69. These values show that the AirBeams demonstrated comparable modest accuracy to the reference instruments at both collocation sites. Multi-variate linear regression was performed at T Street using the same four meteorological variables as at Del Paso Manor, also shown in Table 2. The two most explanatory variables from the multilinear regression analysis were the dew point and relative humidity. At both Del Paso Manor and T street, among the four models with two variables of regression, dew point and relative humidity had the highest adjusted R^2^ values, and the lowest *p*-values (*p*-values less than 10^-16^ when using hourly data, and *p*-values less than 0.005 when using daily averaged data). However, the specific regression coefficients were quite different at the two sites. This is unlikely to be due to the differences in meteorology between the two sites, since temperature, RH, and wind speed were highly correlated (R^2^ values = 0.96, 0.87, and 0.86 respectively) with similar distributions, indicating that meteorology is fairly homogeneous throughout Sacramento. The precision on the BAM is less than that for FRM measurements, roughly 22% [33], so the differences in the regression result may be due more to uncertainties in the operation and precision of the comparison instruments, rather than true meteorological differences. 

Based on these findings, and considering that the AirBeam sensor uses an optical technique, it is likely that hygroscopic aerosol growth played a role in systematically biasing AirBeam measurements in this study to report higher concentration values relative to the values reported by regulatory monitors. The uptake of water vapor from hygroscopic aerosols, such as nitrate and highly oxidized organics can lead to an increase in particle size, and thus to greater scattering efficiency per unit mass concentration [34]. The BAM accounts for aerosol-bound water by heating the incoming stream to evaporate the water [35,36]. The AirBeam sensor, however, does not heat the incoming stream, and so particles enlarged with water scatter more light into the detector, leading to a positively biased measurement. This has also been observed in other optical particle counter-based sensors [32]. Both high relative humidity and high dew point could contribute to this effect, leading to more flux of water to the particles. The measurements during the study period may be especially sensitive to this impact because of the high frequency of saturated air masses in wintertime Sacramento. 

### 3.4. Inter-Community Variability of PM

Corrected AirBeam values were used to examine inter-community variability of PM in Sacramento during the study period. Figure 4 shows a time series of AirBeam measurements from a representative sensor in each of the six communities, the difference between the PM in each community from the median across all six communities, and wind speed and direction for the period from 15 December 2016, to 5 January 2017. This period included holiday periods and the day with the maximum measured concentration during the study, 23 December. A clear diurnal periodicity was found during the study, reflected in both AirBeam and BAM measurements (shown later in Figure 5), with the highest PM concentrations occurring at night. AirBeam values across the community sites typically converge at low PM levels during the day when winds are stronger and the atmospheric boundary layer is higher, and diverge overnight when concentrations are higher. The correlation of pollution hour-by-hour across the sensors suggests that common emission sources influence the entirety of the study domain, with overnight variations among sites, due to the localized influence of sources, such as residential biomass burning and local wind patterns. On average, PM is relatively homogeneous across the study domain, but short-term spikes in PM were common at night-time in most communities, highlighting the localized impact of PM during nighttime conditions. The spatial variability of PM is quantified using statistical measures later in this section. The lower boundary layer height, higher atmospheric stability, and higher emissions due to wintertime heating fuel consumption at night contribute to these PM events, especially when wind speeds are low.

Using periods when all six selected AirBeams in each neighborhood had available data, the hourly-averaged PM_2.5_ concentrations measured from the AirBeams and the two BAMs is shown in Figure 5. These measurements all reflect the same temporal distribution, with higher PM_2.5_ levels during the late evening and night, and levels subsiding over the course of the morning and afternoon. The concentrations start to increase around 4:00 p.m. and rise throughout the evening, likely due to the emissions from rush-hour traffic and residential biomass burning. The measurements presented in Figure 5 also provide a way to examine the spatial variability of PM_2.5_. PM at Del Paso Manor is the highest among the six communities, similar to BC measurements reported in Brown et al. [27].

The distances between AirBeam site pairs during the study period are shown in Table 3. To assess spatial variability, the pairwise statistical measures of correlation coefficients and COD were computed among the AirBeams from the 15 community sites. These results are shown in Table 4 and Table 5 and are derived from both hourly measurements and daily average measurements. The pairwise statistics derived from daily average values show higher correlations compared to hourly values, because the diurnal variability is not present. As expected, these comparisons show a statistically significant relationship to distance, with higher R^2^ and lower COD values between closer AirBeams. For two AirBeams within a 5 km distance, very high correlations are seen with hourly/daily R^2^ values above 0.90/0.95 and with hourly/daily COD values being below 0.20/0.15. Over larger distances, the correlations remain fairly high, indicating modest spatial heterogeneity in PM: the low correlation limit (for AirBeams over a large distance) for hourly/daily R^2^ values is 0.67/0.81 and for hourly/daily COD values is 0.33/0.22. However, these represent outliers from Table 4 and Table 5. A COD value of 0.2 is typically used as a threshold for heterogeneity [29,30], so for the study period, hourly PM_2.5_ was marginally heterogeneous and daily PM_2.5_ was almost always homogeneous within the 14 km by 16 km study domain.

For comparison, Zikova et al. [8] carried out a similar analysis using sensors deployed at 27 sites in Monroe County, NY, over the months from October to April. Compared to Monroe County, correlations, from this study, show a much more homogeneous PM environment, with higher R^2^ values and lower COD values. In Monroe County, pollution levels were much lower overall (with median PM_2.5_ around 3 µg m^−3^), and showed a temporal pattern with higher PM_2.5_ in the daytime, with the transportation sector being a major contributor. PM_2.5_ loading was significantly more homogeneous in Sacramento because pollution is driven by diffuse night-time emissions during stagnant, low-wind conditions in the wintertime, and by the topography of Sacramento Valley, which traps pollutants in the domain.

A Student’s t-test comparison of the measurements from all EJ communities versus all non-EJ communities found no statistically significant difference in the PM_2.5_ levels (*p* value = 0.238). The PM_2.5_ distributions from a representative sensor in each of the six communities outlined in Figure 1 and Figure 4 are shown as a box plot in Figure 6 for the study period. Figure 6 shows that the differences in the distributions of PM_2.5_ are modest among the six communities, with the six communities having similar inter-quartile ranges. The non-parametric Pairwise Wilcoxon Rank comparison was applied to the nine pairwise cases between EJ versus non-EJ sites, in order to assess whether the mean ranks differ. This test showed that for eight cases, there was no statistically significant difference between the means (*p* value > 0.68); however, for the T Street versus South Sacramento comparison, there was a statistically significant difference in the means (*p* value = 0.00046), with the difference in the means being 1.5 µg m^-3^. Appendix A shows the results of the Pairwise Wilcoxon Rank comparison for all pairwise EJ versus non-EJ AirBeam comparisons. These examinations provide evidence indicating that these communities face comparable degrees of exposure to PM_2.5_ concentrations. This analysis and the observed distributions in Figure 6 reflect the fairly homogeneous spatial distribution of PM_2.5_.

### 3.5. Comparison of Measurements to Wintertime Emissions Inventory

The sensor network, spanning across six communities in Sacramento, provides an opportunity to assess the relationship between ambient air pollution and the PM_2.5_ emissions inventory. The AirBeam PM_2.5_ concentrations exhibited statistically significant spatial clustering (Global Moran’s I statistic of 1.0, *p* < 0.005). Higher PM_2.5_ concentrations were measured in the northeast section of the study area (Arden, Del Paso), while lower concentrations were measured south of the American River (South Sacramento, T Street) and in the western portions of South Natomas (Coroval). This general pattern is also seen in the emissions inventory data (see Appendix A for maps). The relationship between the measured concentrations of PM_2.5_ and the emissions inventory (Figure 7) showed a higher R^2^ for weekend days (0.76) than weekdays (0.46). Weekend days showed more variability in measured PM_2.5_ across sites (Range = 5.8 µg m^−3^) than weekdays (Range = 4.2 µg m^−3^). The larger differences between sites on weekend days may account for the stronger observed relationship with the emissions inventory on those days. The modestly high correlation indicates that overall, the EI appears to capture the spatial variability in PM emissions in each grid cell. This indicates that the EI tends to accurately reflect the relative amount of PM_2.5_ emissions in each grid cell, particularly on weekends. The two EI grid cells that had the largest residuals for weekdays were the cells containing the Tristan (in South Sacramento community) and Coroval (in South Natomas community) sites. For these grid cells, the EI may not fully represent emissions, and the location of the monitors may not be completely representative of the entire grid cell, or the PM_2.5_ concentrations in these locations may be affected by factors not captured in the EI, such as secondary aerosol formation and transport. We found that grid cells with higher emissions have higher average PM_2.5_ concentrations, whereas grid cells with lower emission tend to have lower PM_2.5_ concentrations and greater PM_2.5_ variability, similar to the findings of Mohan et al. [24].

As described in Section 3.4, PM_2.5_ has modest heterogeneity among the six communities in the wintertime, where concentrations are typically low during the day and high at night. As seen in Brown et al. [27], as well as in the emissions inventory, residential biomass burning is a large source of PM_2.5_ in Sacramento during the winter, leading to the high nighttime PM_2.5_ concentrations. The emissions inventory and sensor observations are in agreement that the communities with the highest PM_2.5_ concentrations for the six communities in the study are the Del Paso Manor and Arden communities. Other communities in Sacramento may have higher concentrations, but were not among the targeted communities for this study. While, the emissions inventory shows good agreement with measured spatial variability in PM_2.5_ seasonally, while measured PM_2.5_ concentrations were higher in other neighborhoods on some evenings, contributing to the modest heterogeneity observed from the pairwise statistical comparisons in Table 4 and Table 5. 

## 4. Conclusions

In this study located in Sacramento, California, the low-cost AirBeam PM_2.5_ sensors were found to have robust precision with consistent biases among individual AirBeams that were correctable using results from two collocated measurement periods. The AirBeam demonstrated modest accuracy in comparison to BAM and FRM regulatory-grade monitors, with significant bias from the sensor under high dew point and relative humidity conditions. This is likely because of the hygroscopic aerosol growth, under such conditions, that the FEM/FRM measurements can account for, but the AirBeam sensor does not. The AirBeam was shown to be sufficiently precise to make an estimate of spatial variability across multiple communities in Sacramento. Measurements from the AirBeams converged during cleaner periods, with more variable spatial gradients as well as hot spots during periods of elevated PM_2.5_ overnight. Pairwise statistical metrics show that PM_2.5_ levels were mostly homogeneous throughout the study domain for the study period. A gradient of higher PM_2.5_ levels to the northeast of the study domain was observed and corresponded with similar gradients in the gridded emissions inventory. The study design and methodology, that were used to evaluate the AirBeam sensors in this study, could also be applied to other sensor models and other pollutants. While, current sensor technology does not have a regulatory impact due to their limitations, this study demonstrates the value of sensor measurements used to examine spatial and temporal variability of air pollution among communities. In order to utilize sensor measurements for these applications, a careful methodology must be employed with regard to quality control, sensor characterization, and bias correction, so that the data quality of sensor measurements is commensurate with their application. 

## Figures and Tables

**Figure 1 sensors-19-04701-f001:**
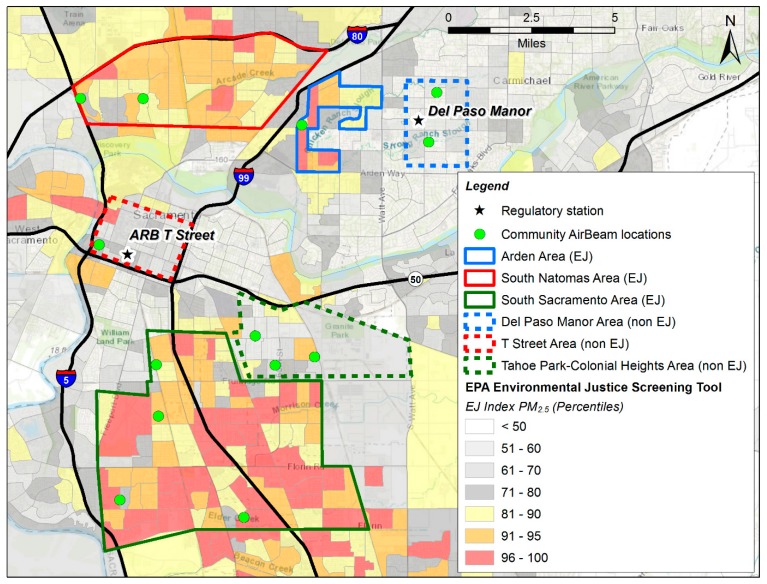
Map of Sacramento study domain showing AirBeam network stations and the two regulatory stations. Community boundaries for EJ and non-EJ communities are shown. Areas shaded yellow, orange and red have a higher EJ Index percentile. Meteorology stations are collocated at the Del Paso Manor and ARB T street sites.

**Figure 2 sensors-19-04701-f002:**
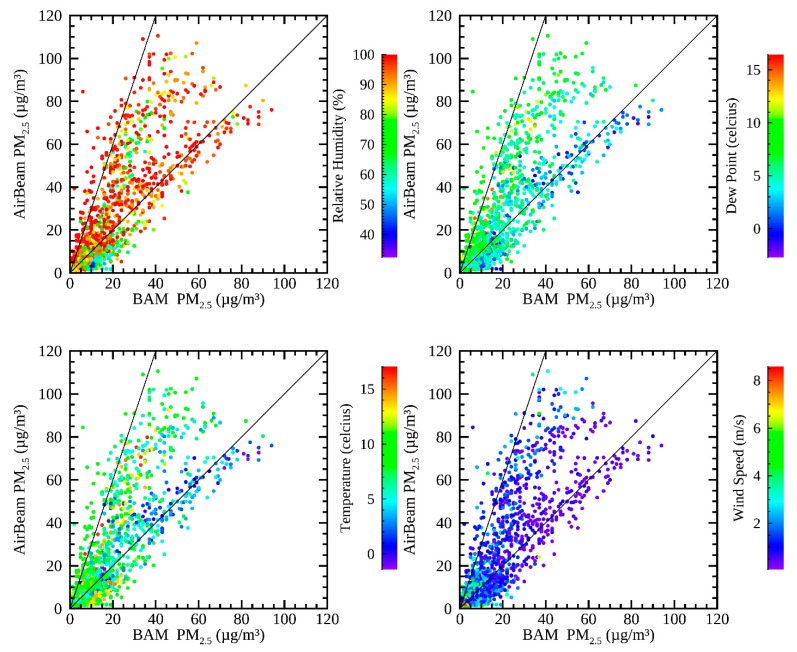
Hourly measurements of PM_2.5_ from the AirBeam compared to the BAM at Del Paso Manor during December 2016-January 2017. The data are color-coded to (**top left**) relative humidity, (**top right**) dew point, (**bottom left**) temperature, and (**bottom right**) wind speed. 1:1 and 3:1 lines are shown for reference.

**Figure 3 sensors-19-04701-f003:**
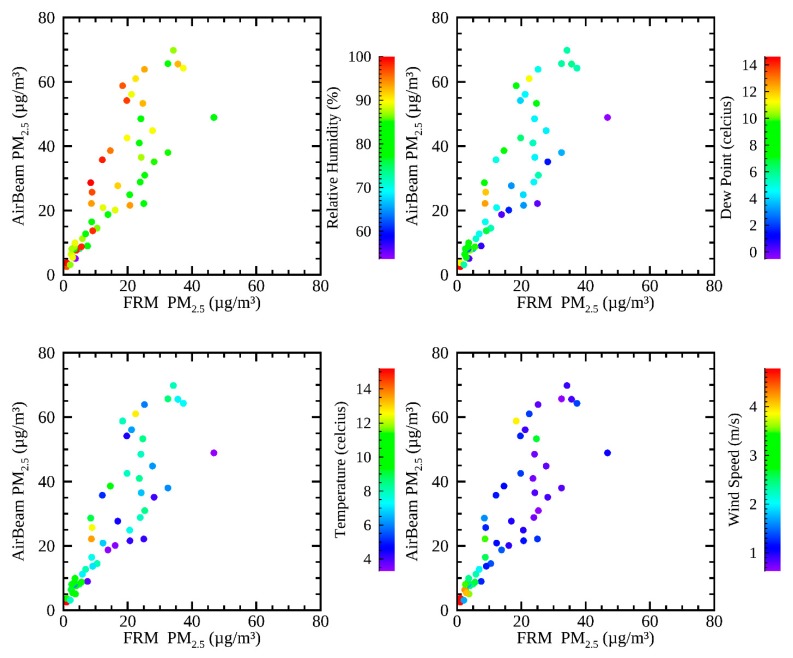
Daily average measurements of PM_2.5_ from the Del Paso Manor AirBeam compared to the FRM. The data are color-coded to (**top left**) relative humidity, (**top right**) dew point, (**bottom left**) temperature, and (**bottom right**) wind speed.

**Figure 4 sensors-19-04701-f004:**
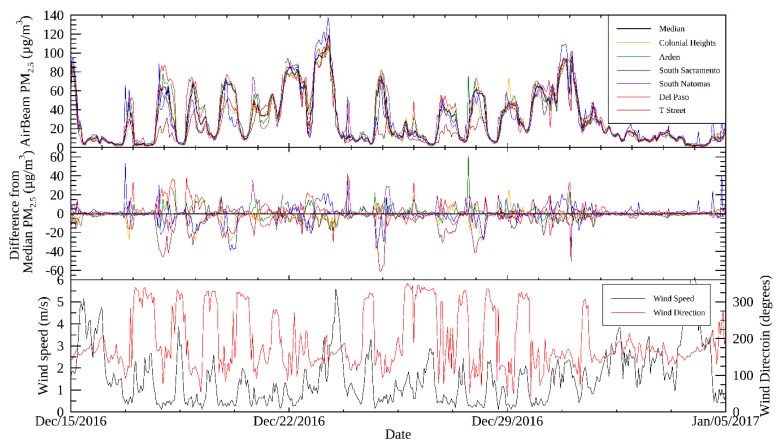
Hourly AirBeam PM_2.5_ measurements from the six communities (top), difference from median AirBeam concentration across the six communities (middle), and wind speed and wind direction measurements measured at Del Paso Manor (bottom) from December 15, 2016, to January 05, 2017. Colonial Heights, Del Paso Manor, and T Street are non-EJ areas, while Arden, South Sacramento, and South Natomas are EJ areas.

**Figure 5 sensors-19-04701-f005:**
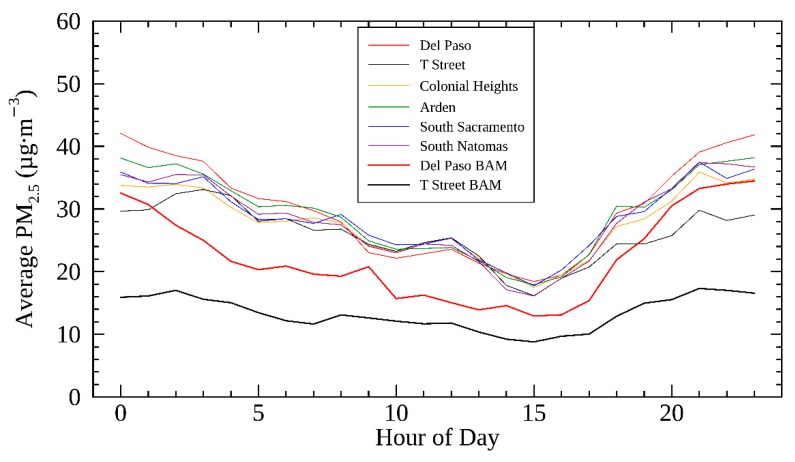
Temporal distribution of two BAM monitors and corrected AirBeam PM_2.5_ measurements from the six communities in the study.

**Figure 6 sensors-19-04701-f006:**
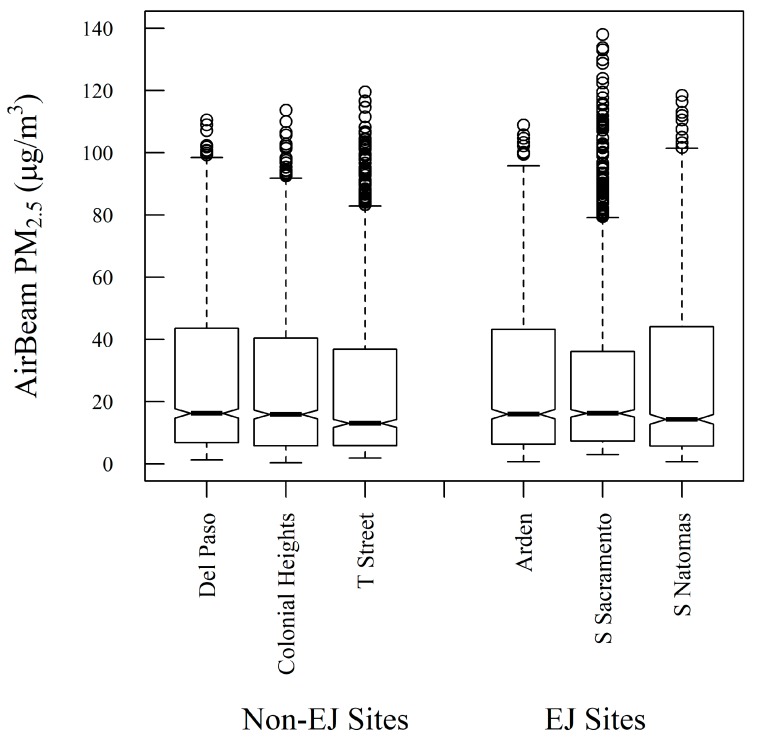
Distribution of corrected AirBeam PM_2.5_ measurements from six communities. The center of the boxplot represents the median value, with 95% confidence interval at the notches. The box cutoffs are the inter-quantile ranges (IQRs), the whiskers represent 1.5 × IQR, and the remaining points in the distribution are plotted individually.

**Figure 7 sensors-19-04701-f007:**
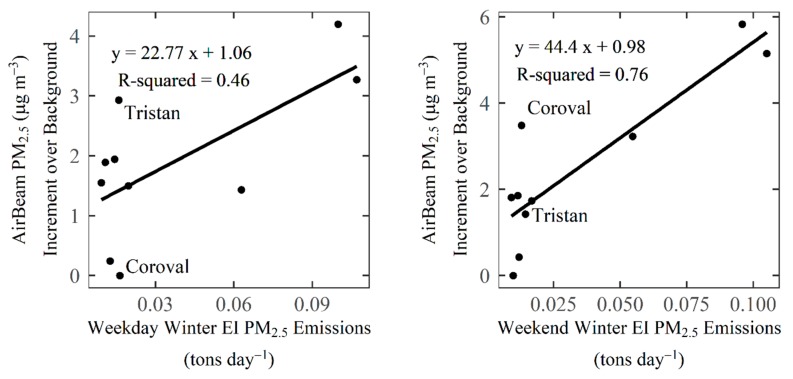
Comparison of gridded winter PM_2.5_ emissions and average AirBeam PM_2.5_ concentrations in the grid cell from December 2016 through January 2017 for weekdays (**left**) and weekends (**right**).

**Table 1 sensors-19-04701-t001:** Precision statistics using collocated measurements from the pre- and post-study periods for Pearson correlation coefficient (R2) values, root mean squared error (RMSE), linear regression intercepts and slopes for the pre and post-study periods. The normalization correction factor was taken to be the linear regression using the average slope and intercept.

	Pre-Study					Post-Study					Correction Factor	
AirBeam Name	N (hours of valid data)	R^2^ vs. AirBeam Means	RMSE vs. AirBeam Means (μg/m^3^)	Slope of regression	Intercept of regression (μg/m^3^)	N (hours of valid data)	R^2^ vs. AirBeam Means	RMSE vs. AirBeam Means (μg/m^3^)	Slope	Intercept (μg/m^3^)	Slope: Average slope	Intercept: Average Intercept (μg/m^3^)
13th Ave	47	0.99	0.38	0.91	0.03	470	0.99	0.69	0.96	−0.19	0.94	−0.08
24th Ave	47	0.99	0.47	0.93	−0.53	469	0.99	0.70	0.96	−0.49	0.94	−0.51
64th St	152	0.99	0.92	1.01	0.93	467	0.99	0.53	1.06	0.09	1.03	0.51
ARB T St 2	146	0.99	1.61	0.81	−0.15	467	0.99	1.02	0.75	0.07	0.78	−0.04
ARB T St 3	150	0.99	1.11	0.95	1.19	467	0.99	0.26	1.02	0.15	0.98	0.67
Alderwood	150	0.99	1.23	0.88	1.51	465	0.99	0.34	0.95	0.09	0.92	0.80
Coroval	138	0.99	0.92	1.15	0.16	378	0.99	0.58	1.26	−0.70	1.20	−0.27
Del Paso 2	152	0.99	1.51	0.80	2.64	465	0.99	1.09	0.87	1.37	0.83	2.00
Del Paso 3	152	0.99	0.81	1.10	−0.53	468	0.99	0.59	1.19	−0.78	1.15	−0.66
Darwin St	88	0.99	0.82	0.99	0.55	466	0.99	0.36	1.04	−0.22	1.01	0.16
Henrietta Dr	47	0.99	0.60	1.55	−1.17	291	0.99	0.59	1.73	−0.51	1.64	−0.84
Socorro Way	88	0.99	0.88	0.96	0.74	464	0.99	0.42	1.00	−0.22	0.98	0.26
Tristan Cir	138	0.98	2.55	0.97	−2.95	466	0.98	1.52	0.94	−1.00	0.95	−1.98
Wyman	47	0.99	0.88	1.10	1.17	467	0.99	1.14	1.07	1.24	1.08	1.21
79th St	62	0.99	0.69	0.93	−0.39	466	0.99	0.65	0.84	−0.71	0.89	−0.55
ARB T St	146	0.99	0.87	1.13	−1.67	392	0.99	0.43	1.06	−0.34	1.10	−1.01
Del Paso	152	0.99	1.17	1.13	0.09	150	0.99	0.43	1.06	0.44	1.13	0.09
Hermosa St	47	0.99	0.74	0.85	−0.30	291	0.99	0.55	0.83	−0.16	0.85	−0.30
T St Tier 3	152	0.98	2.45	1.08	−3.20	291	0.99	0.49	0.97	−0.30	1.08	−3.20

**Table 2 sensors-19-04701-t002:** Multi-linear regressions from the study period using collocated measurements at Del Paso Manor and T Street. Rows show the initial Pearson correlation coefficient (R^2^) between one AirBeam and the regulatory monitor, and the improvement with meteorological explanatory variables. REG indicates a regulatory monitor: 24-h filter-based Federal Reference Method sampler (FRM) for the third column and beta attenuation monitors (BAM) for the other four columns.

Variables of Regression	Hourly BAM vs. AirBeam: Adjusted R^2^	Daily Average BAM vs. AirBeam: Adjusted R^2^	Daily FRM vs. AirBeam: Adjusted R^2^	Hourly BAM vs. AirBeam: Adjusted R^2^	Daily Average BAM vs. AirBeam: Adjusted R^2^
**Monitoring Site**	Del Paso Manor	Del Paso Manor	Del Paso Manor	T Street	T Street
**Initial R^2^**	0.601	0.573	0.716	0.684	0.746
**REG + Temp**	0.604	0.596	0.738	0.686	0.747
**REG + Dew**	0.623	0.641	0.767	0.706	0.776
**REG + RH**	0.617	0.647	0.759	0.703	0.800
**REG + WS**	0.609	0.567	0.716	0.686	0.758
**REG + Temp + Dew + RH + WS**	0.648	0.651	0.762	0.715	0.804
**REG + quadratic (Temp, Dew, RH, WS)**	0.732	0.830	0.883	0.867	0.932

**Table 3 sensors-19-04701-t003:** Matrix of the distances between the 15 AirBeam site pairs during the study period. AirBeam sites are grouped by community. Distances are in kilometers.

Distances	Community	Darwin	Alder	Del Paso	Wyman	Coroval	Socorro	24th Ave	Henrietta	Hermosa	Tristan Cir	ARB T St	Tst Tier 3	13th Ave	64th St.	79th St.
Darwin	Arden	0.0														
Alder	Del Paso	4.8	0.0													
Del Paso	Del Paso	4.4	0.9	0.0												
Wyman	Del Paso	5.3	1.9	1.2	0.0											
Coroval	South Natomas	8.4	13.3	12.8	13.5	0.0										
Socorro	South Natomas	6.1	10.9	10.5	11.1	2.4	0.0									
24th Ave	South Sacramento	8.1	11.6	11.7	12.8	7.5	6.9	0.0								
Henrietta	South Sacramento	15.7	17.9	18.3	19.5	15.2	15.2	8.3	0.0							
Hermosa	South Sacramento	14.3	15.7	16.2	17.4	15.6	15.1	8.2	3.3	0.0						
Tristan	South Sacramento	15.0	15.8	16.4	17.6	17.0	16.3	9.5	4.7	1.7	0.0					
ARB T St	T St	8.2	12.2	12.1	13.2	6.1	5.9	1.5	9.3	9.5	10.9	0.0				
TstTier 3	T St	8.9	13.1	13.0	14.0	5.6	5.8	2.5	9.7	10.2	11.7	1.1	0.0			
13th Ave	Tahoe Park	8.2	9.8	10.2	11.5	11.1	9.9	4.3	8.0	6.2	6.9	5.7	6.8	0.0		
64th St.	Tahoe Park	9.1	10.2	10.7	12.0	12.5	11.2	5.5	7.8	5.5	5.9	7.0	8.1	1.3	0.0	
79th St.	Tahoe Park	8.8	9.2	9.8	11.0	13.2	11.7	6.7	9.1	6.6	6.6	8.1	9.2	2.4	1.5	0.0

**Table 4 sensors-19-04701-t004:** Matrix showing pairwise Pearson coefficient of determination (R^2^) values from the 15 AirBeam sites during the study period. AirBeam sites are grouped by community. R^2^ values are derived from hourly measurements (bottom left half), and daily average measurements (top right half).

R^2^	Community	Darwin	Alder	Del Paso	Wyman	Coroval	Socorro Way	24th Ave	Henrietta	Hermosa	Tristan Cir	ARB T St	Tst Tier 3	13th Ave	64th St.	79th St.
Darwin	Arden	1	0.99	0.89	0.98	0.98	0.92	0.9	0.83	0.84	0.9	0.85	0.81	0.94	0.94	0.91
Alder	Del Paso	0.96	1	0.88	0.98	0.98	0.92	0.9	0.83	0.85	0.9	0.85	0.81	0.95	0.95	0.92
Del Paso	Del Paso	0.73	0.72	1	0.87	0.93	0.97	0.97	0.91	0.94	0.93	0.95	0.96	0.93	0.94	0.91
Wyman	Del Paso	0.94	0.94	0.71	1	0.97	0.91	0.89	0.83	0.83	0.9	0.85	0.81	0.93	0.93	0.9
Coroval	South Natomas	0.93	0.92	0.82	0.9	1	0.97	0.94	0.89	0.88	0.92	0.88	0.86	0.96	0.96	0.92
Socorro	South Natomas	0.81	0.81	0.92	0.79	0.89	1	0.96	0.91	0.92	0.93	0.91	0.91	0.95	0.95	0.89
24th Ave	South Sacramento	0.79	0.8	0.89	0.77	0.86	0.89	1	0.94	0.97	0.98	0.97	0.96	0.97	0.98	0.94
Henrietta	South Sacramento	0.74	0.74	0.84	0.73	0.81	0.84	0.89	1	0.9	0.9	0.91	0.92	0.9	0.92	0.87
Hermosa	South Sacramento	0.71	0.73	0.83	0.71	0.78	0.81	0.91	0.81	1	0.96	0.96	0.96	0.93	0.94	0.92
Tristan	South Sacramento	0.78	0.8	0.79	0.77	0.81	0.81	0.91	0.81	0.86	1	0.96	0.95	0.97	0.97	0.95
ARB T St	T St	0.72	0.72	0.9	0.7	0.8	0.85	0.94	0.85	0.9	0.87	1	0.98	0.94	0.95	0.93
TstTier 3	T St	0.68	0.69	0.88	0.67	0.77	0.83	0.9	0.84	0.87	0.84	0.95	1	0.91	0.92	0.92
13th Ave	Tahoe Park	0.87	0.88	0.82	0.84	0.9	0.87	0.9	0.83	0.83	0.89	0.84	0.79	1	0.99	0.96
64th St.	Tahoe Park	0.87	0.88	0.82	0.85	0.9	0.86	0.92	0.84	0.86	0.9	0.86	0.83	0.95	1	0.97
79th St.	Tahoe Park	0.82	0.84	0.76	0.79	0.84	0.77	0.86	0.75	0.83	0.87	0.82	0.8	0.91	0.94	1

**Table 5 sensors-19-04701-t005:** Matrix showing pairwise coefficient of divergence (COD) values from the 15 AirBeam sites during the study period. AirBeam sites are grouped by community. COD values are derived from hourly measurements (bottom left half), and 24-h average measurements (top right half).

COD	Community	Darwin	Alder	Del Paso	Wyman	Coroval	Socorro Way	24th Ave	Henrietta	Hermosa	Tristan Cir	ARB T	Tst Tier 3	13th Ave	64th St.	79th St.
Darwin	Arden	0	0.07	0.07	0.08	0.16	0.09	0.11	0.11	0.2	0.14	0.18	0.2	0.11	0.08	0.12
Alder	Del Paso	0.16	0	0.06	0.09	0.19	0.12	0.15	0.15	0.21	0.15	0.21	0.22	0.13	0.09	0.13
Del Paso	Del Paso	0.14	0.14	0	0.08	0.18	0.12	0.14	0.14	0.2	0.13	0.2	0.21	0.11	0.11	0.13
Wyman	Del Paso	0.18	0.15	0.17	0	0.18	0.14	0.14	0.14	0.2	0.14	0.2	0.2	0.13	0.12	0.13
Coroval	South Natomas	0.23	0.28	0.25	0.3	0	0.11	0.1	0.13	0.16	0.15	0.11	0.12	0.16	0.14	0.14
Socorro	South Natomas	0.17	0.23	0.21	0.25	0.16	0	0.09	0.11	0.18	0.15	0.15	0.17	0.12	0.1	0.13
24th Ave	South Sacramento	0.18	0.24	0.21	0.26	0.18	0.17	0	0.08	0.14	0.12	0.09	0.13	0.12	0.09	0.09
Henrietta	South Sacramento	0.21	0.26	0.23	0.27	0.2	0.19	0.14	0	0.16	0.13	0.13	0.15	0.13	0.11	0.11
Hermosa	South Sacramento	0.24	0.28	0.25	0.29	0.22	0.23	0.17	0.2	0	0.12	0.15	0.13	0.18	0.18	0.15
Tristan	South Sacramento	0.23	0.27	0.22	0.27	0.24	0.24	0.17	0.2	0.2	0	0.16	0.13	0.11	0.14	0.11
ARBT	T St	0.23	0.27	0.25	0.29	0.17	0.19	0.14	0.19	0.19	0.22	0	0.12	0.18	0.14	0.14
TstTier 3	T St	0.28	0.33	0.28	0.33	0.22	0.27	0.2	0.23	0.23	0.18	0.21	0	0.17	0.19	0.13
13th Ave	Tahoe Park	0.21	0.23	0.22	0.26	0.25	0.22	0.2	0.23	0.25	0.23	0.25	0.28	0	0.11	0.11
64th St.	Tahoe Park	0.15	0.18	0.19	0.22	0.23	0.17	0.16	0.19	0.22	0.22	0.2	0.29	0.19	0	0.09
79th St.	Tahoe Park	0.18	0.21	0.19	0.22	0.23	0.21	0.17	0.21	0.2	0.17	0.21	0.21	0.16	0.14	0

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
