# Peer review of "Measuring Spatial and Temporal PM2.5 Variations in Sacramento, California, Communities Using a Network of Low-Cost Sensors"

_sensors, 2019, doi:10.3390/s19214701_

Round 1

Reviewer 1 Report

This work offers some important results that can be useful when low cost sensor technology is considered for concentration measurements of PM.

The method, comparisons and results are acceptable in their presented form. However, in Section 3.5 the authors attempt to compare concentration values with emissions values from EI. This approach is not scientifically sound and the conclusions drawn from the comparison do not add value to the paper. The EI should be used as input to an air quality model; the calculated by the model concentrations can then be compared with the measured concentrations. Therefore, it is advised to exclude this section from the manuscript and revise the "conclusion" accordingly.

The rest of the paper is sufficient for publication.

Author Response

The authors wish to thank the reviewer for the helpful comments. These comments were carefully considered, changes have been adopted, and this process has strengthened the draft.

The authors have decided to retain Section 3.5 and believe that the results do add value to the paper. Several changes were adopted in order to clarify this analysis, put the results into context, and make sure that results were correctly stated and interpreted.

First, it is acknowledged that we are not doing an apples to apples comparison of concentrations to concentrations. The authors recognize the well-established method of comparing model concentrations to measured concentrations. Instead, we present a semi-qualitative direct comparison of emissions and measured concentrations from a network of sensors – a novel approach that used gridded emissions and average measured concentrations within each grid cell.

This was clarified in the introduction with the addition of the text: “In this study, measurements from a network of AirBeam sensors are compared to a gridded emissions inventory to examine whether emission hotspots result in elevated PM2.5 concentrations detectable by sensors. While directly comparing emissions to concentrations neglects processes such as secondary aerosol formation, it provides a way to examine the spatial variability of emissions and concentrations similar to methods used by Mohan et al.” This text complements the text in Section 3.5, which states that the factors of transport and secondary aerosol formation are not reflected at the level of emissions.

Second, references to other scientific studies were added to put this analysis into context. In particular, the study from Mohan et al. provides a direct comparison to this analysis, comparing gridded emissions inventory to measures concentrations. Text was added in Section 3.5 to compare with the Mohan study: “We find that grid cells with higher emissions have higher average PM2.5 concentrations, whereas grid cells with lower emission tend to have lower PM2.5 concentrations and greater PM2.5 variability, similar to the findings of Mohan et al.”

In conclusion, this comment was carefully considered, and while our study does not employ a full photochemical model, we still believe retaining Section 3.5 adds value to the paper, within proper context. We wish to retain Section 3.5 because it presents a novel approach to assess the spatial variability of measured PM2.5 concentrations from the sensor network within the scope of this work, even though the comparison of emissions and concentrations does not reflect processes such as transport and secondary aerosol formation.

Reviewer 2 Report

In my opinion, the work is interesting, scientifically sounding and sufficiently original. I believe that it could be accepted for publication after the following minor remarks have been addressed.

Abstract. “2.5” for PM2.5 should be changed from superscript to subscript.

Keywords. I suggest to change the last keyword to simply “Network design”.

In my opinion, description of the study area as given at the end of Introduction (markedly, LL96-103: “Sacramento is located… wood burning”) is out of context. I rather entirely move it to a new subsection of “2. Methods” such as, for example, “2.1 Study area” (or something).

In the map of Fig. 1, location of the two meteorological stations is missing: are they co-located with the regulatory BAM and FRM air quality stations?

L119. I suppose a parenthesis is missing after “0.75”. In addition, please check that it is “derived PM2.5” rather than “derived PM2”.

L149. I suppose “are” is missing before “connected”.

L251. Please, rephrase the sentence “The average of…”.

LL297-298. Since previously detailed as Eq. (2) at L206, I believe that it is useless to repeat the formula for corrected AirBeam herein. Rather, a mere citation of Eq. (2) should be enough.

Table 1 could be improved by including a new row for the heading, by grouping columns 2 to 6 under “Pre-study”, columns 7 to 11 under “Post-study”, and columns 12 and 13 under “Correction Factor”. In addition, units in headings for columns 4, 6, 9, 11, and 13 should also be reported.

Please, consider if moving Table 3 to the Supplementary material could be preferable.

L417. I am not fully convinced that citation of Ref. [39] is appropriate. So, please check.

L479. The reported Ref. [8] could be simply cited within the text as “Zikova et al. ]8]”.

Table S1. In Column 3, correct “1 ug/m3” to “10 ug/m3”.

While comparing PM2.5 concentrations and PM2.5 gridded emissions (section 3.5), I believe that it was not sufficiently clarified that PM2.5 concentrations are daily averages (see, for example, Figs. 7, S2, and S3).  

Author Response

The authors wish to thank the reviewer for the helpful comments. These comments were carefully considered, changes have been adopted, and this process has strengthened the draft.

This correction has been made. This suggestion is helpful; change has been made. This suggestion is helpful; description of study area has been moved to Methods Section 2.2, Study Design. Clarification has been added to Figure 1 caption (also stated in Section 2.2, Study Design). These two corrections have been made. This correction has been made. Sentence has been rephrased and clarified. This correction has been made. This suggestion is helpful; change has been made. We considered moving Table 3 to supplemental material. We have decided to retain Table 3 in the main text. The reason we chose to do so is that in Tables 4 and 5, the spatial statistics are only meaningful within the context of the distances shown in Table 3. Keeping Table 3 in the main text, with its identical matrix organization to Tables 4 and 5, allows for quick comparison and interpretation of Tables 4 and 5. This suggestion is helpful; the erroneous reference has been removed. This correction has been made. This correction has been made. Clarification has been added to make clear the time period of averaging.
